# Vitamin D Deficiency in Obese Children Is Associated with Some Metabolic Syndrome Components, but Not with Metabolic Syndrome Itself

**DOI:** 10.3390/metabo13080914

**Published:** 2023-08-04

**Authors:** Jagoda Hofman-Hutna, Michał Hutny, Edyta Matusik, Magdalena Olszanecka-Glinianowicz, Pawel Matusik

**Affiliations:** 1Scientific Society of Medical Students, Faculty of Medical Sciences in Katowice, Medical University of Silesia, 40-055 Katowice, Poland; 2Department of Rehabilitation, Faculty of Health Sciences in Katowice, Medical University of Silesia, 40-055 Katowice, Poland; 3Unit of Public Health and Obesity, Department of Pathophysiology, Faculty of Medical Sciences in Katowice, Medical University of Silesia, 40-055 Katowice, Poland; 4Department of Paediatrics, Paediatric Obesity and Metabolic Bone Diseases, Faculty of Medical Sciences in Katowice, Medical University of Silesia, 40-055 Katowice, Poland

**Keywords:** vitamin D, calcifediol, 25OHD, metabolic syndrome, HDL, adiponectin, TG

## Abstract

Vitamin D deficiency in children is a common nutritional issue in many populations worldwide, associated not only with skeletal malformations but, as recent studies suggest, also with the development of obesity and metabolic syndrome. The aim of this observational study was to assess the nutritional status of vitamin D in a group of Polish children with obesity and different grades of metabolic syndrome, with a consequent analysis of the correlation between vitamin D levels and the components of metabolic syndrome. For that purpose, the group of 78 participants (mean age: 14.18 ± 2.67 years) was recruited and further grouped in relation to vitamin D status into two groups of children with and without vitamin D deficiency. The biochemical parameters associated with obesity as well as anthropometric measures were assessed and analysed in search of significant differences between the groups. In the current group of children with obesity and vitamin D deficiency, HDL (45.00 ± 9.29) and adiponectin (7.21 ± 1.64) were found to be significantly lower than in their peers without vitamin D deficiency, whereas W/HtR (0.60 ± 0.04) and TG (171.31 ± 80.75) levels proved to be significantly higher. Body composition analysis using bioelectrical impedance returned no significant findings. The above findings suggest that vitamin D deficiency may influence lipid and glucose metabolism in children, leading to the development of abnormalities characteristic of the metabolic syndrome. A W/HtR parameter was shown to be a sensitive marker of abdominal obesity, which might provide an important means of assessing the correlation between vitamin D and this type of obesity. Independently, vitamin D deficiency may also influence the endocrinological function of adipose tissue, leading to lower concentrations of adiponectin. These in turn presented a linear correlation with the high results of the OGTT in the second hour of the test, hinting at its potential role in the pathophysiology of insulin resistance.

## 1. Introduction

Vitamin D is involved in the regulation of various processes in the human body, not only in skeletal tissue but—as it was discovered in the last 20 years—also in adipose tissue and many others. Its active form is calcitriol, also known as 1,25-dihydroxycholecalciferol, which is a product of renal and hepatic conversion of cholecalciferol. Cholecalciferol, or vitamin D3, is synthesised in the skin after exposure to UVB light. Deficiency of vitamin D, defined as a serum 25-hydroxyvitamin D (calcifediol, 25OHD) concentration below 20 ng/mL [1], gives rise to comorbidities associated with pathological skeletal development. It might result from low exposure to sunlight, inadequate vitamin D supply in the diet, or renal or hepatic diseases [2].

Metabolic syndrome (MetS) is not a disease itself but rather a combination of risk factors that together contribute to the development of cardiovascular diseases and type 2 diabetes mellitus. In its pathophysiology, the most crucial factor is a long-term positive energy balance. It consists of such features as insulin resistance, high blood glucose level (≥100 mg/dL), dyslipidaemia (TG ≥ 1.69 mmol/L or treatment of high TG; HDL-C < 1.03 mmol/L for men or <1.29 mmol/L for women or HDL treatment), hypertension (≥130/85 mmHg or treatment of hypertension), and high waist circumference (≥102 cm for men or ≥88 cm for women). In order to diagnose a MetS, at least 3 of the 5 above-mentioned features must be found [3].

Previous studies suggested that MetS and comorbidities of obesity are associated with vitamin D deficiency [4,5,6,7,8]. Unrelated to factors such as age and race, vitamin D seems to influence glucose metabolism, insulin resistance, and the inflammatory response, as shown by the positive correlation between levels of vitamin D, leptin, and adiponectin and the negative correlation between vitamin D and the homeostatic model assessment insulin resistance score (HOMA-IR) or waist circumference [9,10]. By modulating levels of adipokines, low vitamin D status may also stimulate the development of cardiovascular diseases (CVD) [10,11].

In this study, an observational analysis was performed on a group of 79 children and adolescents in terms of the correlation between 25OHD levels and anthropometric measures of obesity, levels of adiponectin, markers of MetS, as well as MetS itself. The aim of this study was to evaluate the previously reported results and confront them with the results obtained from the Polish paediatric group.

## 2. Materials and Methods

### 2.1. Study Participants

The participants of the study were recruited from the population of patients with obesity. Exclusion criteria comprise obesity secondary to endocrine disorders and treatment with medicaments reportedly influencing either metabolic conditions or vitamin D and calcium metabolism. As a result, a final total of 79 children were included in the study group, with a mean age of 14.18 ± 2.67 years.

### 2.2. Anthropometric Parameters

All participants underwent the measurement of BMI-z score, fat percentage (FAT%), fat free mass (FFM%), and assessment of severe obesity (Severe ob). The latter was defined in accordance with the International Obesity Task Force (IOTF) BMI cut-off tables [12], translating the measured paediatric BMI to the corresponding values for adults. The cut-off point for severe obesity was a BMI equivalent to the BMI of an adult equal to 35.0. In all but one patient, the waist-to-hip ratio (WHR) and waist-to-height ratio (W/HtR) were measured. The predicted muscle mass (PMM%) was measured for 70 patients. The body composition parameters were assessed using bioelectrical impedance analysis.

### 2.3. Biochemical Assay

The standard metabolic biochemical markers were measured from the venous blood samples in the course of routine clinical laboratory testing. The assessment of the standard marker of vitamin D status—25OHD—was conducted using enzyme-linked immunosorbent assay (ELISA) kits (Immundiagnostik AG, Bensheim, Germany). As indicated in the product manual, the fresh blood samples were centrifuged within an hour of sampling, followed by the collection of serum, which was consequently frozen and stored at a temperature of −20 °C.

### 2.4. Statistical Analyses

For the purpose of statistical analysis, the STATISTICA software was used (13.3.721.1 StatSoft Polska Sp. z.o.o., Cracow, Poland).

For each group (Vitamin D deficient (VDD) vs. Vitamin D sufficient (VDS); patients with MetS vs. patients without MetS), the mean values of anthropometric and biochemical parameters were calculated, as well as standard deviations.

A student’s *t*-test for independent samples was used to compare the anthropometric and biochemical parameters between groups of patients with and without vitamin D deficiency. The same test was used for the comparison of 25OHD values between groups of patients with and without MetS. The equality of variances of parameters between groups was calculated using the Kolmogorov-Smirnov test for equality of two variances. Correlations were measured using the Pearson correlation coefficient. The significance threshold for each test was established as a *p*-value of <0.05.

### 2.5. Ethical Considerations

The study was approved by the Ethics Committee of the Medical University of Silesia (Approval No. KNW/0022/KB1/131/15). All participants and their carers gave informed consent. Patient rights were also approved according to the Helsinki Declaration.

## 3. Results

The summary of anthropometric and biochemical measures of patients, divided into VDD (61 patients) and VDS (18 patients) groups, is presented below in Table 1.

The group consisted of 35 male and 44 female patients, with a mean age of 14.18 ± 2.67. There was no significant difference in BMI-z between the groups; however, the patients in the VDD group showed significantly higher W/HtR (mean: 0.64), which is one of the measures of MetS [13]. This parameter also correlated significantly (r = −0.24; *p* < 0.05) with the levels of 25OHD, as presented below in Figure 1. It must be noted that the groups were significantly different in terms of age (*p* < 0.05). The values of vitamin D were also significantly higher (*p* < 0.01) in female patients than in male patients.

For a summary of biochemical parameters in groups, see Table 1. The groups differed significantly (*p* < 0.0001) in terms of 25OHD levels—the mean value for the VDD group was 12.66 ± 4.76, classifying it as a vitamin D deficiency, whereas the mean value for the VDS group was 27.78 ± 5.59, which is a suboptimal value. Some of the parameters of lipid metabolism were significantly different between the groups—the VDD group presented lower values of HDL (*p* < 0.05) and higher values of TG (*p* < 0.02) than the VDS group. Values of these parameters (mean Chol = 178.43; mean LDL = 99.90; mean HDL = 45.00; mean TG = 171.31) suggest the risk of dyslipidaemia. No significant difference between the groups was observed in terms of levels of aminotransferases. Although the levels of adiponectin were significantly lower in VDD patients (*p* < 0.02), the groups differed also in terms of the variances of this parameter, which indicated using the Welch t test instead of the student’s *t* test for this parameter.

The results of the current study did not prove any significant difference between the groups in terms of glucose metabolism and insulin resistance, as evidenced by the comparison of levels of glucose and insulin at points 0 and 120 min of the OGTT. HOMA-IR scores were also similar in groups (mean VDD vs. mean VDS: 4.68 vs. 3.39). Nonetheless, a significant linear correlation between 25OHD and glu120 (r = −0.28; *p* < 0.05) in the general population of the study was calculated (Figure 2).

The patients were also divided in respect of being diagnosed with MetS (33 patients) or not (45 patients). Twelve patients met only one criterion for MetS. The highest number presented two features of MetS, but only the patients with three or four features of MetS, respectively sixteen and seventeen patients, met the requirements for the diagnosis of MetS. All of them were compared in terms of 25OHD levels, as shown below in Table 2.

Although the difference between the groups in terms of the levels of 25OHD was not statistically significant, it did differ between the groups (mean 15.43 vs. mean 16.44). Moreover, the variances of this parameter were significantly different in the MetS and non-MetS groups. The mean values of 25OHD were, however, in the range of vitamin D deficiency for both groups.

## 4. Discussion

This study presents an analysis of anthropometric measures of obesity, and biomarkers of metabolic syndrome in Polish paediatric groups of vitamin-D deficient and vitamin-D sufficient children and adolescents. W/HtR and TG levels were significantly higher in the VDD group, whereas HDL and adiponectin levels were significantly lower. Levels of other studied parameters did not present any significant differences. The levels of these parameters, as well as the significance of differences in their terms between the studied groups, were summarised for the results of the present study and studies to which the present ones were compared in Table 3 below. In some previous studies, the data concerning mean values and standard deviations of W/HtR and adiponectin were missing. The lack of available data is represented with a “-“ mark.

### 4.1. Anthropometric Measures

Insufficient levels of 25OHD were shown to be associated with higher levels of W/HtR [17,18], but, as reported in some sources, with lesser strength than other adiposity anthropometric parameters [18]. Although it may result from nutritional rickets and growth retardation related to low levels of 25OHD and calcium [19], it was also shown to result from the increased weight of children with VDD, as reflected by a high BMI [17]. Contrary to the results of studies that focused on parameters like BMI, WC, and WHR without examining W/HtR, it was the latter parameter that proved to be significantly different between the groups in this study, whereas WHR and BMI-z scores did not. The significance of the difference in W/HtR values was also higher (*p* = 0.02) than in the previous report (*p* < 0.05) [17]. This parameter is considered useful in assessing abdominal obesity and cardiovascular disease in children and adults; in the latter group, it is also reported to be superior to other anthropometric parameters such as WHR and BMI [20,21].

### 4.2. Lipid Metabolism

The lipid profiles of children with and without VDD differed significantly in terms of HDL (*p* < 0.05) and TG (*p* < 0.02) levels. Low values of HDL cholesterol are associated with the development of cardiovascular diseases [21,22], as well as type 2 diabetes mellitus [23], which underlines the importance of HDL as a MetS component. Interestingly, research on African patients showed that low HDL is the most frequent MetS component in this population [24]. The correlation between 25OHD levels and HDL-c is a matter of discussion; the results of studies show both the presence and the lack of such a correlation. Research conducted on Korean and Iranian children without obesity [6,14], as well as Tunisian children with obesity [7], showed a lack of significant differences in HDL levels between the VDD and VDS groups. Moreover, supplementation of 25OHD in African-American adolescents led to no significant difference in HDL levels between VDD and VDS adolescents, which is consistent with the results of studies on adolescents with obesity [8,15]. Results of another study showed similar differences in HDL levels, although after adjusting them for ethnicity, socioeconomic status, WC, and BMI, these observations were no longer significant [5]. On the contrary, a positive correlation between HDL and 25OHD levels (r = −0.306; *p* < 0.001) was found in a previous study on 114 Turkish children and adolescents. The results of the present study did not show any significant correlation between these two parameters (r = 0.02; *p* > 0.05), though the HDL levels did differ significantly between the VDD and VDS groups (*p* < 0.05).

Low levels of HDL with simultaneous high levels of TG are described as atherogenic dyslipidaemia, which is associated not only with the development of atherosclerosis but also insulin resistance [25]. Hypertriglyceridemia influences the mean arterial pressure, leading to arterial hypertension as well as other cardiovascular diseases [26]. Contrary to HDL, high TG levels have been widely recognised as a MetS component that is significantly correlated with 25OHD nutritional status in children and adolescents with and without obesity [4,6,7,14]. The difference in TG levels between previously studied populations of children with deficient and sufficient 25OHD levels remained statistically significant after adjustment for BMI, age, and gender [16]. However, in adolescents with severe obesity and in children with or without obesity, no statistical difference in TG levels between VDD and VDS groups was observed [5,8]. Supplementation of 25OHD also did not lead to any significant improvement of TG levels in groups of African-American adolescents with obesity, independent of its success in reinstating 25OHD sufficiency [15]. Levels of TG measured in the population of the current study differed significantly (*p* < 0.02) between children with and without VDD and were more significant than in most studies on associations between 25OHD and TG levels [7,14,16].

### 4.3. Adiponectin

Obesity is associated with the inflammatory process of adipose tissue in the whole body, leading to decreased levels of adiponectin [27], which is a component of the metabolic syndrome. The Jackson Heart Study showed that higher values of adiponectin in male African Americans correlate with a lower risk of hypertension [28]. Although lower adiponectin levels are considered to be associated with the development of insulin resistance and type 2 diabetes mellitus, studies examining this link returned ambiguous results [29,30]. Recently, an adiponectin-involving parameter—the leptin/adiponectin ratio—was suggested as a measure reflecting insulin resistance [31]. In the studied group of patients, the leptin/adiponectin ratio was not significantly different between the VDD and VDS groups, nor did it correlate with levels of 25OHD. It did, however, present a significant correlation (r = −0.24; *p* < 0.05) with values of blood sugar in the second hour of OGTT, which suggests its association with the development of insulin resistance. Elevated levels of adiponectin also correlated significantly with impairment of left atrial functions and consequently with the frequency of atrial fibrillation in the examined populations of patients with diabetes and hypertension [32]. In the previous studies, the adiponectin levels differed significantly between the groups of children with and without vitamin D deficiency (*p* < 0.001) [4], even after adjustment for age and gender (*p* = 0.02) [5]. Supplementation of 25OHD in African-American adolescents with obesity did not result in a significant difference in adiponectin levels between the VDD and VDS groups [15]. In the present study, significantly lower (*p* < 0.02) concentrations of serum adiponectin were found in the VDD group than in the VDS group (mean: 7.21 ± 1.64 vs. 8.84 ± 3.95). The correlation between vitamin D levels and adiponectin levels (r = 0.17) did not reach significance (*p* > 0.05).

### 4.4. Glucose Metabolism

According to the results of previous studies, children whose vitamin D levels are not sufficient in childhood are more likely to be in the later stages of life than those with normal values of 25OHD [33,34], though there are also reports in which this association was not observed [35]. According to the interventional trials, the parameters of glucose metabolism in relation to 25OHD improved after the lifestyle intervention, which included dietary counselling and regular physical activity [33], although the supplementation of vitamin D alone did not result in such an improvement [34]. None of the glucose metabolism-related parameters was significantly different between the VDD and VDS groups in the current study, but differences in levels of some of them—glu120, ins0, and HOMA IR—were approaching significance (respectively: *p* = 0.14; *p* = 0.19; *p* = 0.18). Together with the significant correlation between 25OHD and glu120 levels in the population of the current study, the above observations indicate that hypovitaminosis D does influence glucose metabolism and, as the previous studies suggest, may lead to the development of related comorbidities, such as insulin resistance.

### 4.5. Metabolic Syndrome

Despite the lack of a significant difference (*p* = 0.59) in severity of MetS between the groups of children with or without vitamin D deficiency, the levels of MetS components (W/HtR, HDL, TG, and adiponectin) were shown to be significantly altered between the groups (*p* < 0.03; *p* < 0.04; *p* < 0.02; *p* < 0.02), with increased levels of W/HtR and TG and lowered levels of HDL and adiponectin. Children meeting 3 or 4 MetS criteria had a lower mean 25OHD level (15.43) than those meeting only 1 or 2 MetS criteria (16.44). In either case, however, these mean values were within the range of vitamin D deficiency.

Taking all the above observations into consideration, as well as the fact that differences in many of the examined parameters—BMI-z score, glu120, ins0, and HOMA-IR—were approaching significance, it is clear that the examined children who had an inadequately low level of vitamin D present a wide range of metabolic disorders that, when not addressed properly, might result in the development of severe complications. Observed hypertriglyceridaemia, an increased waist-to-height ratio, and lower levels of HDL and adiponectin indicate that these patients are at high cardiovascular risk, despite the fact that they do not always meet the required criteria for MetS, as both groups of children (MetS and non-MetS) showed mean values of 25OHD falling into the category of vitamin D deficiency. Considering the ambiguity of the results of previous studies examining the role of vitamin D in MetS as well as the prominent influence of 25OHD on the metabolic condition of patients, it may be reasonable to reassess the applicability of MetS criteria in the cases of patients with hypovitaminosis D.

To the best of the authors’ knowledge, this study is the first to describe the correlations between vitamin D deficiency and the components of MetS in the Polish paediatric population, with simultaneous comparison with the results of studies on groups of children from other regions of the world.

### 4.6. Limitations

The main limitation of this study is its relatively small population and the unbalanced number of patients enrolled in the VDD and VDS. Studies on larger populations are needed to confirm the findings and generalise them to broader populations. The groups of children with and without vitamin D deficiency were adequately matched for age, severe obesity, and BMI-z score; however, they were significantly different in terms of gender (*p* < 0.005). Mean vitamin D concentrations also differed between the overall groups of boys and girls enrolled in the study, though the significance of this difference could not be examined statistically due to the imbalance of groups in terms of gender ratio. An issue of gender differences in vitamin D concentrations is difficult to address, as the results of studies on this matter are ambiguous [36,37], and there are no separate reference ranges of serum vitamin D concentrations for males and females. Due to the enrollment system based on the patients attending control visits at the clinical centre, other confounding factors affecting the results, such as diet, physical activity, genetic predisposition, and seasonal variations in vitamin D levels, were not taken into consideration in this study. The latter factor is especially crucial, as the variability of serum vitamin D concentrations between the astronomical winter and summer months is apparent in Poland [38], similarly to the neighbouring countries [39]. As suggested by the results of studies on adults and children, the differences in duration of sun exposure might also be an important factor [40,41].

Many of the observed differences were approaching the significance threshold, which indicates that research on larger groups of patients is required to sufficiently examine the relations between the parameters. Patients were divided into groups characterised by sufficient/deficient levels of vitamin D, whereas it might be useful to use a category of suboptimal vitamin D levels (20–30 ng/mL 25OHD), as the patients with these results could give an insight into the early metabolic changes associated with hypovitaminosis D.

## 5. Conclusions

Observational analysis of correlations between levels of vitamin D and components of MetS in a group of 79 children with obesity and different degrees of MetS showed significant differences in levels of anthropometric and biochemical parameters recognised as important factors of MetS. Children with VDD had significantly lower levels of HDL and adiponectin and significantly higher W/HtR and TG levels than children with sufficient levels of 25OHD. Significant correlations between 25OHD, W/HtR, and glu120 values were observed. However, the obtained results failed to show a significant difference in 25OHD levels between children with and without MetS.

## Figures and Tables

**Figure 1 metabolites-13-00914-f001:**
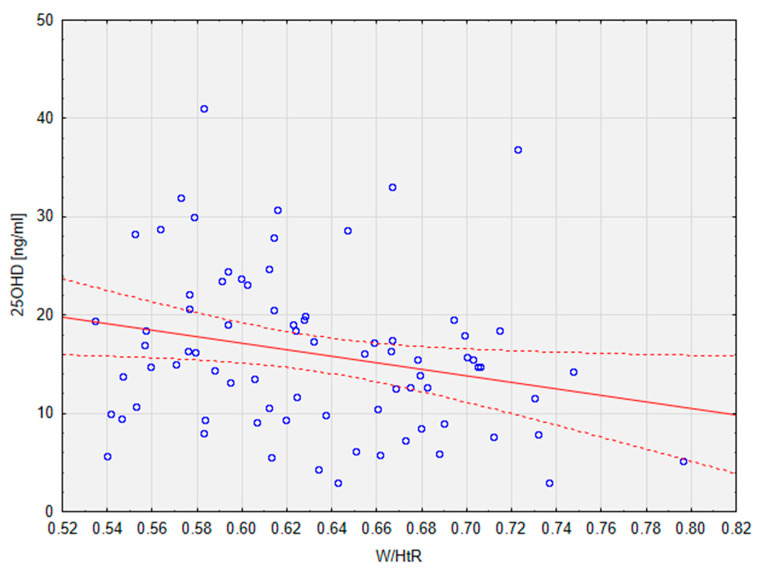
Correlation between 25OHD levels and the W/HtR in the studied population. Dotted line—95% confidence interval. 25OHD, calcifediol, 25-hydroxyvitamin D; W/HtR, Waist to height ratio.

**Figure 2 metabolites-13-00914-f002:**
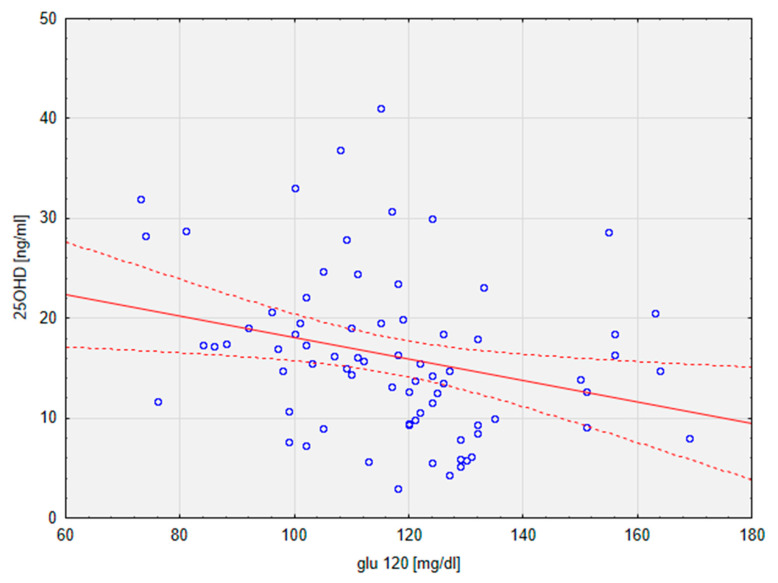
Graph depicting a linear regression of 25OHD on glu120. Dotted line—95% confidence interval. 25OHD, calcifediol, 25-hydroxyvitamin D; glu120, glucose at 2 h of OGTT.

**Table 1 metabolites-13-00914-t001:** Summary of studied parameters in groups of patients with and without vitamin D deficiency.

Parameter	Unit	Mean VDD ± SD VDD (*n* = 61)	Mean VDS ± SD VDS (*n* = 18)	*p*
Age	Years	14.50 ± 2.49	13.04 ± 2.98	<0.05
Gender	M/F	33/28	2/16	<0.005
WHR	-	0.97 ± 0.07	0.95 ± 0.06	NS
W/HtR	-	0.64 ± 0.06	0.60 ± 0.04	<0.03
BMI-z score	-	3.06 ± 0.96	2.71 ± 0.78	NS
Severe ob	-	1.48 ± 0.50	1.33 ± 0.49	NS
FAT%	%	39.78 ± 7.50	37.47 ± 5.15	NS
FFM%	%	60.22 ± 7.50	62.48 ± 5.21	NS
PMM%	%	56.78 ± 7.21	59.40 ± 4.85	NS
Chol	mg/dL	178.43 ± 32.69	184.00 ± 29.42	NS
HDL	mg/dL	45.00 ± 9.29	50.13 ± 6.87	<0.05
LDL	mg/dL	99.90 ± 29.64	103.43 ± 36.39	NS
TG	mg/dL	171.31 ± 80.75	121.76 ± 51.93	<0.05
AlAT	IU/L	40.28 ± 37.42	30.83 ± 23.61	NS
AspAT	IU/L	33.13 ± 20.54	25.63 ± 7.47	NS
glu0	mg/dL	90.41 ± 9.35	89.12 ± 8.35	NS
glu120	mg/dL	119.24 ± 19.57	110.82 ± 24.47	NS
ins0	mIU/L	20.68 ± 16.51	15.27 ± 7.24	NS
ins120	mIU/L	103.83 ± 80.11	97.98 ± 39.88	NS
HOMA IR	-	4.68 ± 3.83	3.39 ± 1.71	NS
IR	-	1.58 ± 0.50	1.53 ± 0.51	NS
25OHD	ng/mL	12.66 ± 4.76	27.78 ± 5.59	<0.0001
Leptin	ng/mL	57.11 ± 45.61	63.52 ± 52.30	NS
Adiponectin	μg/mL	7.21 ± 1.64	8.84 ± 3.95	<0.02
Leptin/Adiponectin ratio	-	8.33 ± 6.72	7.79 ± 6.29	NS

AlAT, alanine aminotransferase; AspAT, asparagine aminotransferase; Chol, total cholesterol; FAT%, fat percentage; FFM%, fat-free mass; glu0, glucose at point 0 min of OGTT; glu120, glucose at point 120 min of OGTT; HDL, high-density lipoprotein cholesterol; HOMA-IR, homeostatic model assessment insulin resistance score; ins0, insulin at point 0 min of OGTT; ins120, insulin at point 120 min of OGTT; IR, insulin resistance; M/F, male to female participant ratio; LDL, low-density lipoprotein cholesterol; *n*, number of patients; NS, not significant; PMM%, predicted muscle mass; SD, standard deviation; TG, triglyceride; VDD, vitamin D deficient group; VDS, vitamin D sufficient group; WHR, waist to hip ratio; W/HtR, waist to height ratio; 25OHD, 25-hydroxyvitamin D.

**Table 2 metabolites-13-00914-t002:** Comparison of calcifediol levels in groups of patients with and without MetS.

Parameter	Mean MetS ± SD MetS	Mean *n*MetS ± SD *n*Mets	*p*
25OHD	15.43 ± 6.19	16.44 ± 9.25	NS

MetS, patients with MetS; *n*, number of patients; *n*MetS, patients without MetS; NS, not significant; 25OHD, 25-hydroxyvitamin D.

**Table 3 metabolites-13-00914-t003:** Comparison of results between the present study and the literature.

Authors; Year	Present Study			Kim et al., 2019 [14]			Magge et al., 2018 [15]			Censani et al., 2018 [16]		
Groups	1. VDD vs. 2. VDS			1. VDD vs. 2. VDS			1. VDD vs. 2. VDS			1. VDD vs. 2. VDS		
	Mean 1	Mean 2	*p*-Value	Mean 1	Mean 2	*p*-Value	Mean 1	Mean 2	*p*-Value	Mean 1	Mean 2	*p*-Value
25OHD (ng/mL)	12.7 ± 4.8	27.8 ± 5.6	<0.0001	13.6 ± 3.6	25.7 ± 4.9	-	12.0	24.1	<0.0001	15.2	27.5	-
W/HtR	0.6 ± 0.1	0.6 ± 0.0	<0.05	-	-	-	-	-	-	-	-	-
TG (mg/dL)	171.3 ± 80.8	121.8 ± 51.9	<0.05	90.3 ± 49.4	74.7 ± 31.0	<0.01	78.3	65.4	NS	130.2	78.9	<0.05
Cholesterol (mg/dL)	178.4 ± 32.7	184.0 ± 29.4	NS	169.0	169.6	NS	155.9	166.0	NS	184.2	158.9	<0.01
LDL (mg/dL)	99.9 ± 29.6	103.4 ± 36.4	NS	95.0	94.7	NS	95.3	103.7	NS	112.5	93.7	<0.05
HDL (mg/dL)	45.0 ± 9.3	50.1 ± 6.9	<0.05	56.6 ± 12.5	59.1 ± 1	NS	43.0	51.0	NS	45.3	50.0	NS
Adiponectin (μg/mL)	7.2 ± 1.6	8.8 ± 4.0	<0.05	-	-	-	3.0	3.3	NS	-	-	-

## Data Availability

Data is available upon request. The data are not publicly available due to the privacy laws of the hospitals in which the patient was treated.

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
