# Peer review of "Vitamin D Deficiency in Obese Children Is Associated with Some Metabolic Syndrome Components, but Not with Metabolic Syndrome Itself"

_metabolites, 2023, doi:10.3390/metabo13080914_

Round 1

Reviewer 1 Report

The findings of this study could have important implications for public health policy and clinical practice. If the link between vitamin D deficiency and metabolic syndrome in children is confirmed in further studies, it could suggest the need for increased efforts to prevent and treat vitamin D deficiency in children, particularly those who are overweight or obese. This could involve measures such as promoting vitamin D-rich foods and increasing exposure to sunlight, as well as providing supplements in cases of deficiency. While this study provides valuable insights into the potential link between vitamin D deficiency, obesity, and metabolic syndrome in children, it also has some shortcomings that should be taken into account.

It is an observational study, which means that it cannot establish causality. While the study found a correlation between vitamin D deficiency and certain markers of metabolic syndrome, it cannot prove that vitamin D deficiency is the cause of these markers. Other factors, such as diet, physical activity, and genetic predisposition, could also play a role.

This study had a relatively small sample size of 78 participants. While the sample size was appropriate for an exploratory study, larger studies would be needed to confirm the findings and generalize them to broader populations.

This study only measured vitamin D levels at one point in time, which may not be representative of the participants' overall vitamin D status. It is possible that some participants had seasonal variations in vitamin D levels or had previously taken supplements that could have influenced their results.

This study did not control for all potential confounding variables that could affect the relationship between vitamin D deficiency and metabolic syndrome.

There are some grammatical mistakes and syntax errors that should be removed through critical revision.

Author Response

Dear Reviewer,

thank you very much for drawing the attention to the points which were not adequately addressed in the current study. It helped enormously to significantly improve the quality of the manuscript. Your remarks, listed in italics, are listed below with an appropriate response following each of them.

It is an observational study, which means that it cannot establish causality. While the study found a correlation between vitamin D deficiency and certain markers of metabolic syndrome, it cannot prove that vitamin D deficiency is the cause of these markers. Other factors, such as diet, physical activity, and genetic predisposition, could also play a role.

This study only measured vitamin D levels at one point in time, which may not be representative of the participants' overall vitamin D status. It is possible that some participants had seasonal variations in vitamin D levels or had previously taken supplements that could have influenced their results.

This study did not control for all potential confounding variables that could affect the relationship between vitamin D deficiency and metabolic syndrome.

As all above remarks concern the same issue – the confounding factors – they are all addressed collectively. A notice of them was taken and implemented into the section 4.6. limitations in lines #318-321, as stated below:

“Due to the enrolment system basing on the patients attending control visits at the clinical centre, the other confounding factors affecting the results, such as diet, physical activity, genetic predisposition and seasonal variations in vitamin D levels were not taken into consideration in this study.”

This study had a relatively small sample size of 78 participants. While the sample size was appropriate for an exploratory study, larger studies would be needed to confirm the findings and generalize them to broader populations.

The remark concerning the sample size was added to the manuscript in lines #308-309, as presented below:

“Studies on larger populations are needed to confirm the findings and generalize them to broader populations.”

Reviewer 2 Report

In this manuscript authors explore the association between VitD and components of METS in obese polish children.

Several concerns to be fully addressed.

1) Table 1, 2 and 3 must be merged in a single table. On first row, please indicate the number of subjets belonging to each groups, i.e. VDD (n= 45).

Some parameters must be better explained, what morbid OB stands for? Furthermore, no information regarding the sex of participants id presented in Table

2) As stated in limitation, the main flaw of this study is the small sample size: However, another critical point is the unbalance of the two grups (VDD (61 patients) and VDS (18 patients) groups that probably influence the final findings. Again, although we are analyzing children, the sex distribution between the two groups may have a significant impact on VitD mean levels and authos must listed this parameter.

3) ViTD levels is influensed the seasonal variations. No information about this issue is available. Please comment this point and include among references the follow important paper: PMID: 32342444

The average quality of english meeds to be improved, in particular in Abstract

Author Response

Dear Reviewer,

we would like to thank you deeply and sincerely for your time and attention given to our manuscript. Your accurate remarks provided us with another perspective to our work, which helped to increase the quality of the article tremendously. We address them in the original order listed below, with an appropriate response to each point.

1) Table 1, 2 and 3 must be merged in a single table. On first row, please indicate the number of subjets belonging to each groups, i.e. VDD (n= 45).

Some parameters must be better explained, what morbid OB stands for? Furthermore, no information regarding the sex of participants id presented in Table

Table 3. cannot be merged with tables 1. and 2., as the tables 1. and 2. compare the groups of vitamin D deficient and vitamin D sufficient children, whereas table 3. compares the patients in relation to their metabolic syndrome status. The table 1. and 2. were merged accordingly together and appropriate information concerning the gender ratio in groups was added.

The morbid obesity was explained according to the definition of IOTF in lines #81-84, as stated below:

“[…] morbid obesity (Morbid ob). The latter was assessed in accordance with International Obesity Task Force (IOTF) BMI cut-off tables12, translating the measured paediatric BMI to the corresponding values for adults. The cut-off point for morbid obesity was BMI equivalent to the BMI of adult equal to 35.0.”

2) As stated in limitation, the main flaw of this study is the small sample size: However, another critical point is the unbalance of the two grups (VDD (61 patients) and VDS (18 patients) groups that probably influence the final findings. Again, although we are analyzing children, the sex distribution between the two groups may have a significant impact on VitD mean levels and authos must listed this parameter.

Respective part of section 4.6. Limitations was changed accordingly in lines #308-310 and #313-318, as stated below:

“The main limitation of this study is its relatively small population and the unbalanced number of patients enrolled in the VDD and VDS. Studies on larger populations are needed to confirm the findings and generalize them to broader populations.”

“Mean vitamin D concentrations also differed between the overall groups of boys and girls enrolled into the study, though the significance of this difference could not be examined statistically due to the imbalance of groups in terms of gender ratio. An issue of gender differences in vitamin D concentrations is difficult to address, as results of studies on this matter are ambiguous37,38 and there are no separate reference ranges of serum vitamin D concentrations for males and females.”

3) ViTD levels is influensed the seasonal variations. No information about this issue is available. Please comment this point and include among references the follow important paper: PMID: 32342444

The issue of seasonal and daily variations is addressed in lines #318-325, as stated below:

“Due to the enrolment system basing on the patients attending control visits at the clinical centre, the other confounding factors affecting the results, such as diet, physical activity, genetic predisposition and seasonal variations in vitamin D levels were not taken into consideration in this study. The latter factor is especially crucial, as the variability of serum vitamin D concentrations between the astronomical winter and summer months is apparent in Poland39, similarly to the neighbouring countries40. As suggested by the results of studies on adults and children, the differences in duration of sun exposure might also be an important factor41,42.”

4) The average quality of english meeds to be improved, in particular in Abstract

The manuscript was proof-read and edited in order to improve its linguistic quality.

Round 2

Reviewer 2 Report

No moire requests

Author Response

Dear Reviewer,

One more time thank you for your valuable comments and remarks.

Authors